# Trying to use temporal and kinematic parameters for the classification in wheelchair badminton

Ilona Alberca[1]*, Bruno Watier[2,3], Félix Chénier[4,5], Florian Brassart[1],
Mélanie Baconnais[6], Bryan Le Toquin[6], Imad Hamri[6], Jean-Marc Vallier[1], Arnaud Faupin[1]

1 Laboratoire J-AP2S, Université de Toulon, La Garde, France, 2 LAAS-CNRS, Université de Toulouse, CNRS, UPS, Toulouse, France, 3 CNRS-AIST JRL (Joint Robotics Laboratory), IRL, National Institute of Advanced Industrial Science and Technology (AIST), Tsukuba, Ibaraki, Japan, 4 Mobility and Adaptive Sports Research Lab, Department of Physical Activity Sciences, Université du Québec à Montréal, Montreal, QC, Canada, 5 Centre for Interdisciplinary Research in Rehabilitation of Greater Montreal, Institut Universitaire sur la Réadaptation en Déficience Physique de Montréal, Montreal, QC, Canada, 6 Institut de Recherche Bio-Médicale et d'Épidémiologie du Sport (IRMES), EA 7329, Institut National du Sport, de l'Expertise et de la Performance (INSEP), Paris, France

* ilona.alberca@univ-tln.fr

## Abstract

### Introduction

This study explores the potential for the temporal and kinematic datas link to propulsion technique and athlete performance collected here to contribute to evidence-based classification for wheelchair badminton athletes.

### Materials and methods

Nineteen experienced wheelchair badminton players underwent propulsion tests with a badminton racket. Wheelchair were equipped with inertial measurement units. The first analysis conducted involved comparing the parameters between class WH1 and WH2. Subsequently, a hierarchical clustering analysis was performed on the parameters with significant differences.

### Results

Regarding propulsion technique parameters, WH1 athletes exhibit a longer braking phase compared to WH2 athletes. Generally, the performance of WH1 athletes is inferior to that of WH2 athletes. Concerning hierarchical clustering analysis, the results reveal the formation of three clusters based on principal components explaining 70% of the variation in the parameters considered in the analysis.

### Conclusion

Thus, the results of this study indicate a longer braking time for WH1 athletes compared to WH2, along with lower overall performance. The clusters results could suggest a potential evolution of the current classification towards three distinct classes of wheelchair

**Data availability statement:** All relevant data are within the manuscript and its Supporting Information files.

**Funding:** This work was supported by a French government grant, managed by the Agence Nationale de la Recherche (ANR) under the "France 2030" program, reference ANR-19-STHP-0005. The funders had no role in study design, data collection and analysis, decision to publish, or preparation of the manuscript.

**Competing interests:** The authors declare no conflict of interest.

badminton players. However, these findings should be interpreted with caution, given that the included performance parameters can be influenced by numerous factors, potentially undermining the robustness of the clustering methodology employed. This study highlights the need to strengthen the current classification process in wheelchair badminton.

## Introduction

Classification of athletes is paramount in wheelchair sports, including badminton, aiming to allocate athletes into appropriate sport classes to minimize the influence of impairment on competition outcomes while prioritizing sporting excellence [1]. As emphasized by Goosey-Tolfrey and Leitch [2], classification reflects the individual's level of disability. Following the functional classification system proposed by Strohkendl in 1982, endorsed by the International Wheelchair Basketball Federation, wheelchair badminton employs a similar approach, assessing athletes' functional abilities for classification purposes [3]. Specifically, they are classified into two classes: WH1 and WH2. WH1 athletes are manual wheelchair users with abdominal and lower limb paralysis, while WH2 participants possess abdominal capabilities but experience lower limb paralysis with partial sensation [4].

Given the novelty of the sport, wheelchair badminton classification was recently established by the International Paralympic Committee Athlete Classification Code in 2015. The classification process is initiated by determining eligibility based on a Minimal Impairment Criteria (MIC), as described by the BWF [3]. Once this step is completed, a physical evaluation is conducted, which includes a manual muscle test [5] and/or a joint mobility test [5]. The ASIA score should be used for athletes with spinal cord injuries [3,6]. Finally, after the physical evaluation, a technical assessment is carried out during a tournament and a training match, where evaluators are asked to identify the following profiles:

- WH1: "Players exhibit functional limitations based on muscle power and trunk range of motion, and possibly upper limbs, during a match or training session" [3].

- WH2: "Players have a functional limitation based on reduced muscle power or range of motion, requiring the use of mobility aids. Shifting the center of gravity may result in loss of balance, for example, when attempting to pivot or during stop-and-start movements" [3].

This final assessment may include an evaluation of the player's ability to perform specific tasks and activities that are part of wheelchair badminton [3].

However, this classification process does not involve any objective measurement of the athletes' performance capabilities. The International Paralympic Committee (IPC) has established the conceptual framework of evidence-based classification for several years [7]. Recent studies have explored the use of data-driven approaches to improve classification in wheelchair sports. Inertial sensor technology and standardized field tests have been shown to provide objective measures of wheelchair mobility performance, potentially reducing the number of classes in wheelchair basketball [8]. Cluster analysis of isometric strength tests has produced valid classification structures for wheelchair track racing, offering a more transparent and less subjective system [9]. In wheelchair rugby, trunk strength impairment has been correlated with specific performance determinants, and cluster analysis has supported the concept of "natural classes" based on how trunk muscle strength affects activities [10]. These data-driven approaches show promise in enhancing the validity and objectivity of classification in wheelchair sports. Based on the study by Tweedy & Vanlandewijck [7], incorporating such parameters into the evidence-based classification process could correspond to "Step 3b: Develop measures of (determinants of) performance". Notably, the integration of inertial

measurement units (IMUs) has recently facilitated the acquisition of on-field performance data for wheelchair athletes [11]. In a manuscript in pre-print and currently under submission, Alberca et al. [12], recently utilized two IMUs placed on the wheels of the wheelchair to assess diverse performance parameters among wheelchair badminton players during a one-minute field test (forward and backward propulsion test reproducing the movements of wheelchair badminton players).

Hence, the main aim of this study is to explore the potential for the temporal and kinematic data link to propulsion technique and athlete performance collected to contribute to evidence-based classification for wheelchair badminton athletes. To meet this main objective, it will first be necessary to examine whether classification has an impact on these same temporal and kinematic parameters. For individuals with spinal cord injuries, the level of the lesion significantly impacts postural stability and propulsion abilities, particularly in the abdominal region, leading to expected performance differences between WH1 and WH2 athletes, as evidenced by studies showing variations in match intensity and shots played [13–18]. Thus, it can be hypothesized that WH1 athletes with more severe functional limitations will demonstrate temporal and kinematic patterns indicative of lower performance compared to WH2 athletes with lower velocities, acceleration, deceleration and longer sprint time. Additionally, it is hypothesized that the temporal and kinematic parameters measured in this study correctly discriminate the two classes of wheelchair badminton athlete.

## Materials and methods

### Study design

This study aims to explore the possibility that the temporal and kinematic data representative of propulsion technique and athlete performance from this study could serve as evidence for the evidence-based classification. Warmed-up participants performed consecutive forward and backward sprints over 3 m for 1 min to carry out a test as close as possible to match conditions as shown in Fig 1 [12]. The duration of 1 min and the length of 3 m of the test were chosen to correspond to the characteristics of this sport without inducing too intense an effort for the athletes (court size: 3.96 m long and effective playing time: 5.7 min and 7.7 min) [17].

During the experiment, all athletes started from a stationary position at the 3-meter line, demarcated by cones, initiating in forward propulsion. Subsequently, they braked and proceeded in backward propulsion along the same course, repeating this sequence for 1 min. When switching between propulsion directions, athletes were required to pass the large wheels of their wheelchairs beyond the cones at each end of the track. Two trials were conducted for each participant: one with the racket and one without, with the trial order determined randomly [12]. The badminton racket and wheelchair utilized were individualized to each participant and matched those employed in competitions. The athletes' personal wheelchairs featured camber angles ranging from 18° to 20°, with wheel sizes ranging from 24 to 26 inches and a rear anti-tip wheel. Each athlete held the racket on their preferred side, referred to as the racket side. A 5-minute break was kept between each trial. Although the propulsion technique was not prescribed, upon observation, all athletes employed synchronous propulsion.

### Ethics

The data of this study were collected during the French Championships of Nueil-les-Aubiers from 14 to 16 January 2022 and Saint-Orens from 13 to 15 January 2023. The experimental protocol was approved by the Comité d'Ethique pour les Recherches en STAPS (CERSTAPS) from Conseil National des Universités de France [certificate #CERSTAPS **IRB00012476-2021-11-06-274**] filed on February 2021 and accepted on Jun 2021. Participants were

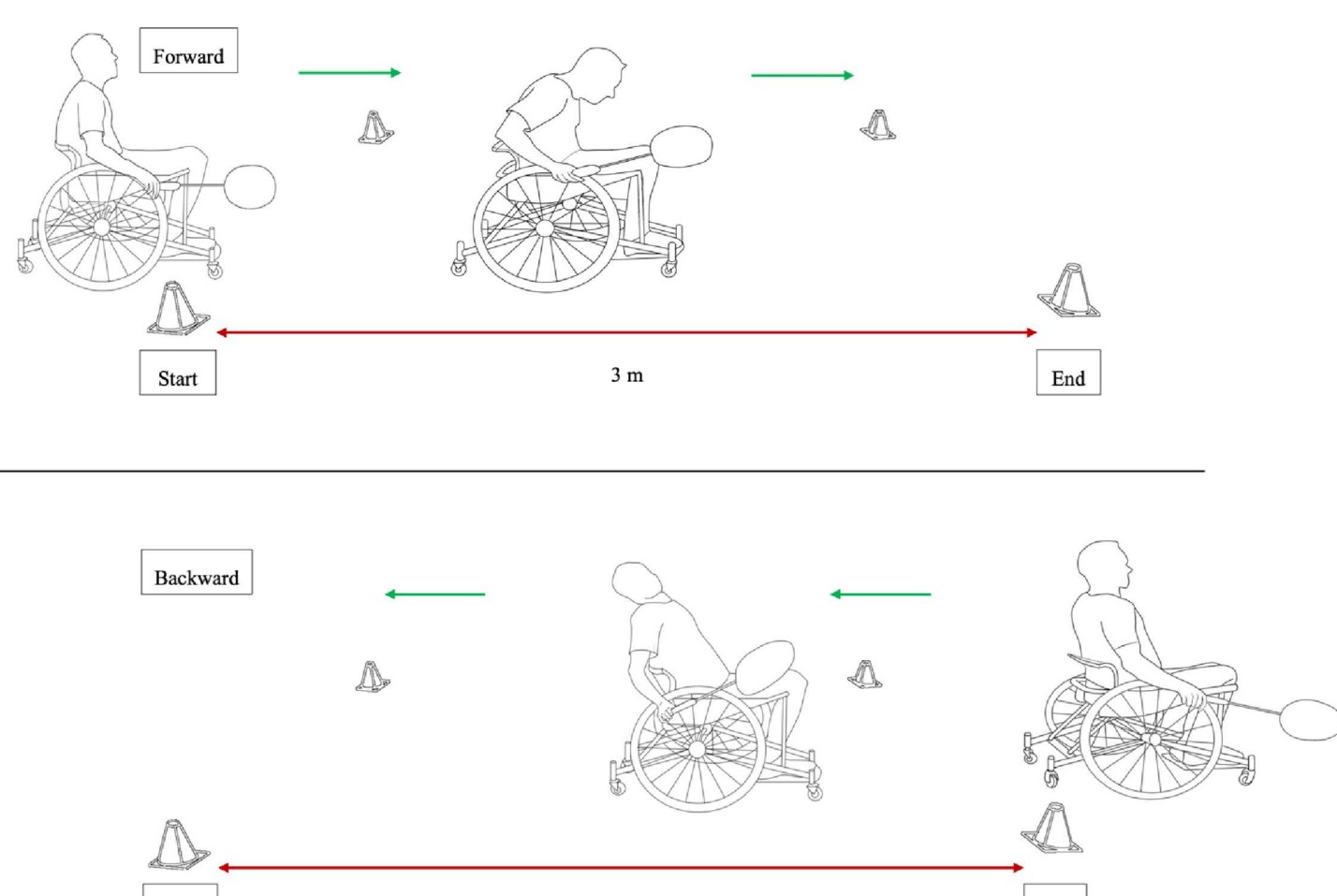

**Fig 1. Schematic diagram of forward/backward propulsion test.**

recruited starting on December 1, 2021, and end on January 10, 2022. All participants have received and signed a written informed consent and information notice.

## Participants

A total of 19 wheelchair badminton athletes was included in this study. Inclusion criteria required participants to be at a national level or higher in wheelchair badminton and have a minimum of one year of experience in playing the sport. Participants were excluded if they reported any pain or injury that could hinder their ability to propel their wheelchair. To determine the minimum sample size required for this study, a statistical power test has been made with de Rhodes et al. [19] as the reference article. The required sample size was estimated at N = 16 participants. Considering this result, a total of N = 19 badminton athletes was included in this analysis. Statistical power testing was performed using G*Power software (G*Power, 2020; g-power.apponic.com). Characteristics of all participants are presented in Table 1.

## Data measurement

Inertial measurement units (IMUs) were used to collect on-field data [11,20–22]. Their wheelchair was equipped bilaterally with two IMUs (128 Hz, 3*3: accelerometer, gyroscope,

**Table 1. Participants' characteristics.**

| | Gender | Class | Age (years) | Body height (cm) | Body mass (kg) | BMI (kg/m²) | Years of practice (years) | Racket side | Health condition | Wheelchair Camber (°) | Wheel size (inch) |
|---|---|---|---|---|---|---|---|---|---|---|---|
| 1 | Female | WH2 | 55 | 162 | 60 | 22,86 | 9 | R | Paraplegia (T12-L1) | 20 | 24 |
| 2 | Female | WH1 | 45 | 165 | 58 | 21,30 | 10 | R | Paraplegia (T6-T8) | 18 | 25 |
| 3 | Male | WH2 | 31 | 180 | 60 | 18,52 | 6 | L | Paraplegia (T12-L1) | 18 | 26 |
| 4 | Male | WH1 | 37 | 176 | 67 | 21,63 | 6 | R | Paraplegia (T7-T8) | 20 | 25 |
| 5 | Male | WH1 | 45 | 158 | 64 | 25,64 | 17 | R | Spinabifida | 18 | 25 |
| 6 | Male | WH2 | 48 | 187 | 75 | 21,45 | 9 | R | Paraplegia (T12) | 20 | 26 |
| 7 | Female | WH1 | 53 | 171 | 68 | 23,26 | 8 | R | Paraplegia (T12-L1) | 20 | 25 |
| 8 | Male | WH1 | 45 | 168 | 71 | 25,16 | 12 | R | Paraplegia (T5-T6) | 20 | 25 |
| 9 | Female | WH1 | 33 | 165 | 60 | 22,04 | 2 | R | Paraplegia (T12-T6) | 20 | 24 |
| 10 | Female | WH2 | 22 | 135 | 43 | 23,59 | 6 | R | Osteogenesis imperfecta | 18 | 24 |
| 11 | Male | WH2 | 38 | 185 | 63 | 18,41 | 2 | R | Paraplegia (T5-T6) | 20 | 25 |
| 12 | Male | WH2 | 44 | 165 | 58 | 21,30 | 9 | R | Paraplegia (T12-L2/L3) | 18 | 25 |
| 13 | Male | WH1 | 40 | 187 | 70 | 20,02 | 5 | R | Paraplegia (T5-T6) | 20 | 25 |
| 14 | Male | WH1 | 49 | 185 | 94 | 27,47 | 5 | R | Paraplegia (T3-T4) | 20 | 25 |
| 15 | Male | WH1 | 52 | 160 | 60 | 23,44 | 3 | L | Poliomielitis | 20 | 25 |
| 16 | Female | WH2 | 41 | 175 | 68 | 22,20 | 9 | R | Incomplete paraplegia (L1-L2) | 18 | 25 |
| 17 | Female | WH2 | 37 | 170 | 60 | 20,76 | 3 | R | Incomplète paraplegia (T12-L1) | 20 | 25 |
| 18 | Female | WH2 | 27 | 156 | 47 | 19,31 | 4 | R | Algoneurodystrophy | 20 | 25 |
| 19 | Male | WH1 | 33 | 178 | 100 | 31,56 | 14 | R | Paraplegia (T6) | 20 | 25 |
| Mean (SD) | | | 40.8 (8.8) | 169.9 (12.7) | 65.6 (13.1) | 38.4 (6.0) | 7.3 (4.0) | | | | |

*SD: standard deviation; BMI: Body Mass Index*

magnetometer, and Bluetooth module, WheelPerf System, AtoutNovation, France). IMUs were placed on each wheel hub, and the gyroscope was used to estimate the direct rotational velocity of the wheel around the z-axis, considering the camber angle of the wheelchair as indicated by Fuss et al. [23]. The z-axes of gyroscopes were placed perpendicularly to the wheel planes [24]. The data were filtered using a Butterworth low-pass filter of order 2 with a cutoff frequency of 4 Hz.

Data were processed using Python 3.11 and Kinetics Toolkit 0.11 [25]. Fig 1 shows an example of propulsion velocity curve for one sprint in forward and backward propulsion. The various phases visible in Fig 1, such as acceleration, deceleration, or the transition phases, were manually marked using events and enabled us to calculate the outcome parameters. All the outcome parameters were calculated for all the 3 meters sprints performed by the athletes.

## Outcome parameters

To meet the objectives of this article, the same propulsion technique and performance parameters as in the study by Alberca et al. [12] were used and are listed in Table 2.

In addition to the definitions shown in Table 2, Fig 2 illustrates the various parameters calculated.

The acceleration and propulsion phase time were calculated only on the beginning of each sprint because it is the only moment when athletes accelerate the most from a stationary position since the wheelchair is stopped and has no velocity. The same reasoning is applied to the deceleration phase time and deceleration. These parameters are only calculated in the end of the sprint since it is the only moment when athletes brake to stop the wheelchair and completely decelerate [12].

**Table 2. Description of the outcome measures.**

| Parameters | Description |
|---|---|
| **Propulsion technique parameters** | |
| Propulsion phase time ($PP_{mean}$) [s] | Time between the sprint start and the first peak velocity |
| Deceleration phase time ($DP_{mean}$) [s] | Time between the last peak velocity and the sprint end |
| **Performance parameters** | |
| Sprint time ($ST_{mean}$) [s] | Sprint time of each direction of propulsion |
| Transition time ($TT_{mean}$) [s] | Time between the end of the deceleration phase and the start of the next sprint |
| Maximum velocity ($V_{max}$) [m/s] | Maximum velocities reached on all sprint |
| Mean velocity ($V_{mean}$) [m/s] | Mean velocities reached on all sprint |
| Peak velocity ($V_{peak}$) [m/s] | First maximum velocity reached during the sprint |
| Acceleration ($A_{mean}$) [m/s²] | Mean acceleration between the sprint start and the first peak velocity |
| Deceleration ($D_{mean}$) [m/s²] | Mean deceleration between the last peak velocity and the sprint end |

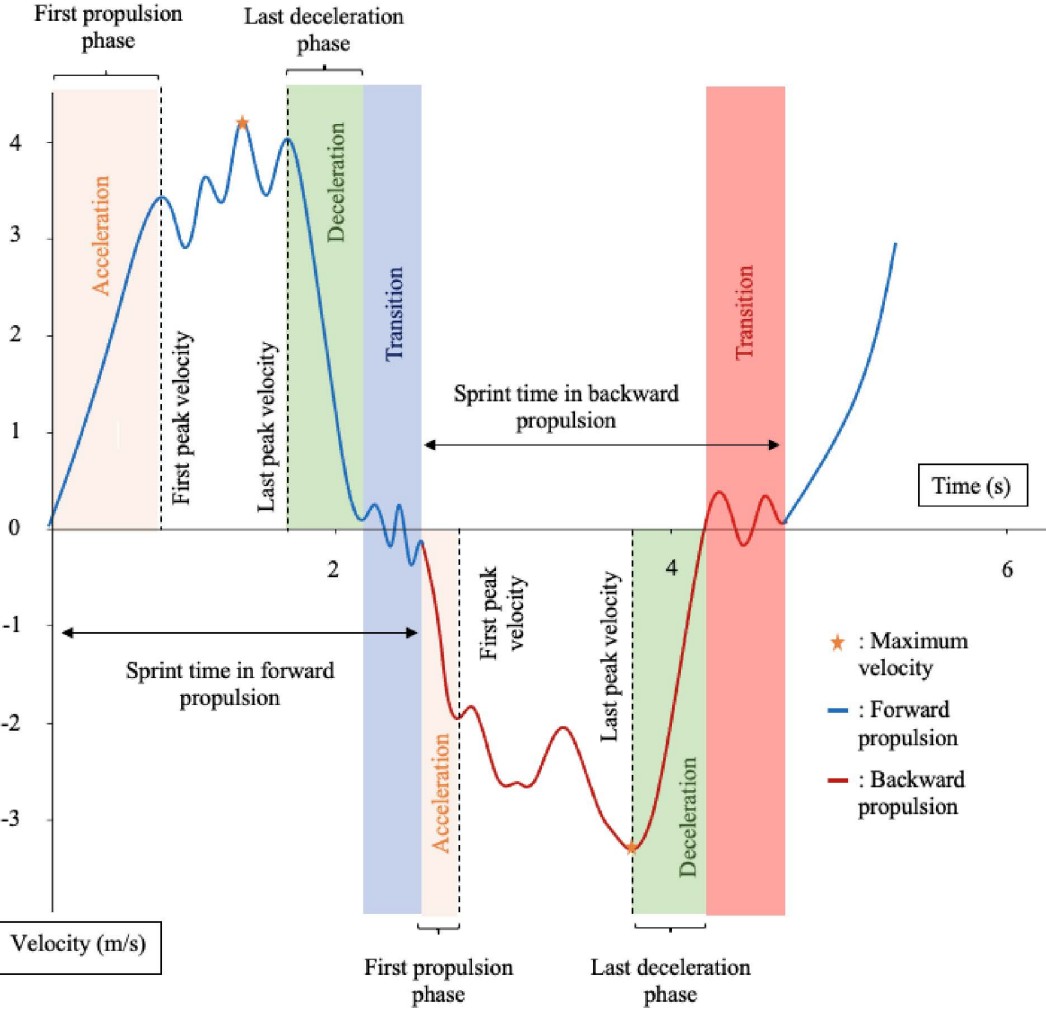

**Fig 2. Example of propulsion velocity curve for one sprint in forward and backward propulsion.**

## Statistical methods

To meet the main objective of our study, the first step is to compare the data of WH1 and WH2 athletes in forward propulsion and in backward propulsion. To do this, the averaged IMU data for the left and right wheel were used, in forward and backward propulsion, for WH1 and WH2 athletes. Normality of the data was tested using a Shapiro-Wilk test, which showed that the data were not normally distributed. Therefore, non-parametric independent Mann-Whitney tests has been chosen for the comparisons. Significance was set at $p < 0.05$.

For each significant difference, the effect size $r$ was calculated using the following equation:

$$r = \frac{Z}{\sqrt{N}}$$

*With Z: statistical result of the statistical test for the parameter under consideration; N: sample size.*

Effect size was interpreted according to [26]: small ($0.10 \leq r \leq 0.29$), moderate ($0.3 \leq r \leq 0.49$), and large ($r \geq 0.5$).

After this initial analysis, the main objective is to explore the possibility that the temporal and kinematic data from this study could serve as evidence for the evidence-based classification of wheelchair badminton athletes. To this end, a principal component analysis (PCA) was performed on parameters showing a significant difference between WH1 and WH2 according to the previous statistical analysis. Propulsion technique parameters and performance parameters with no significant differences between WH1 and WH2 were not considered in the analysis. Then, hierarchical clustering was performed on the coordinates obtained through PCA to identify similarities across classifications. This clustering method is based on iteratively merging data into larger clusters based on their Euclidean distance. In this hierarchical clustering analysis, it is interesting to examine whether the clusters exhibit significant differences among them and to identify on which principal component these differences manifest between the clusters to enhance the understanding of the clusters. To test the hypothesis of a difference between clusters on each PCA component, two non-parametric Kruskal-Wallis tests were performed, as the normality hypothesis was rejected by the Shapiro-Wilk test. Then, pairwise comparisons of clusters on each PCA component were conducted using the Mann-Whitney post-hoc test with Bonferroni correction. Lastly, to understand the distribution of PC classifications within the clusters, a contingency table was created. Results are reported in percentage (%).

## Results

The processed data set used in this article are presented in supporting information file S1 Table.

### Impact of classification on performance

The results of the comparison of data between WH1 and WH2 athletes are presented in Table 3 (a) for forward propulsion and Table 3 (b) for backward propulsion and, in Fig 3.

Regarding forward propulsion, WH1 athletes demonstrate slightly to moderately higher values for $ST_{mean}$ and $DP_{mean}$ compared to athletes in the WH2 class. Conversely, WH1 athletes exhibit slightly to moderately lower values for $V_{max}$, $V_{peak}$, $A_{mean}$, and $D_{mean}$ in comparison to WH2 athletes. No significant difference was found regarding $PP_{mean}$, $TT_{mean}$ and $V_{mean}$ between WH1 and WH2 athletes.

About backward propulsion, like forward propulsion, WH1 athletes exhibit significantly moderately higher values for $DP_{mean}$, $ST_{mean}$ and $TT_{mean}$ compared to WH2 athletes. Conversely, WH1 athletes show significantly lower values for $V_{max}$, $V_{mean}$, $V_{peak}$, and $D_{mean}$ compared to WH2 athletes. No significant difference was found regarding $PP_{mean}$ and $A_{mean}$ between WH1 and WH2 athletes.

**Table 3. Comparison of propulsion technique and performance parameters between the two classes (WH1 and WH2) for the forward propulsion and the backward propulsion.**

| Forward | WH1 | WH2 | Comparison | |
|---|---|---|---|---|
| | Mean(SD) | Mean(SD) | p | r |
| **Propulsion technique parameters** | | | | |
| $PP_{mean}$ (s) | 0.47(±0.17) | 0.44(±0.18) | 0.376 | 0.076 |
| **$DP_{mean}$ (s)** | **0.45(±0.17)** | **0.37(±0.11)** | **0.002*** | **0.263** |
| **Performance parameters** | | | | |
| **$ST_{mean}$ (s)** | **2.19(±0.78)** | **1.98(±0.29)** | **<0.001** | **0.414** |
| $TT_{mean}{}^{1}$ (s) | 0.39(±0.23) | 0.41(±0.28) | 0.806 | 0.021 |
| **$V_{max}$ (m/s)** | **4.31(±0.70)** | **4.71(±0.59)** | **<0.001*** | **0.332** |
| $V_{mean}$ (m/s) | 2.94(±0.53) | 3.05(±0.42) | 0.079 | 0.150 |
| **$V_{peak}$ (m/s)** | **2.98(±0.94)** | **3.15(±0.89)** | **0.015*** | **0.208** |
| **$A_{mean}$ (m/s²)** | **5.36(±1.76)** | **6.08(±1.65)** | **<0.001*** | **0.300** |
| **$D_{mean}$ (m/s²)** | **8.46(±3.56)** | **11.77(±3.80)** | **<0.001*** | **0.482** |
| Backward | WH1 | WH2 | Comparison | |
| | Mean(SD) | Mean(SD) | p | r |
| **Propulsion technique parameters** | | | | |
| $PP_{mean}$ (s) | 0.43(±0.15) | 0.43(±0.15) | 0.483 | 0.060 |
| **$DP_{mean}$ (s)** | **0.43(±0.15)** | **0.35(±0.11)** | **<0.001*** | **0.323** |
| **Performance parameters** | | | | |
| **$ST_{mean}$ (s)** | **2.42(±0.63)** | **2.27(±0.75)** | **<0.001** | **0.453** |
| **$TT_{mean}{}^{2}$ (s)** | **0.43(±0.25)** | **0.41(±0.28)** | **<0.001** | **0.344** |
| **$V_{max}$ (m/s)** | **3.84(±0.67)** | **4.17(±0.54)** | **<0.001*** | **0.347** |
| **$V_{mean}$ (m/s)** | **2.64(±0.36)** | **2.83(±0.31)** | **<0.001*** | **0.321** |
| **$V_{peak}$ (m/s)** | **2.66(±0.51)** | **3.05(±0.81)** | **<0.001*** | **0.292** |
| $A_{mean}$ (m/s²) | 5.98(±1.63) | 6.26(±1.55) | 0.126 | 0.131 |
| **$D_{mean}$ (m/s²)** | **7.61(±2.63)** | **11.09(±3.46)** | **<0.001*** | **0.589** |

*SD: standard deviation; p: p-value fixed at 0.05; r: effect size for the significant difference; Bold values indicate significant values; 1: transition time from forward propulsion to backward propulsion; 2: transition time from backward propulsion to forward propulsion; $PP_{mean}$: propulsion phase time on the first push; $DP_{mean}$: deleration phase time on the last push; $ST_{mean}$: sprint time; $TT_{mean}$: transition time; $V_{max}$: maximum velocity; $V_{mean}$: mean velocity; $V_{peak}$: peak velocity on the first push; $A_{mean}$: acceleration on the first push; $D_{mean}$: deceleration on the last push.*

In the initial analysis, significant differences in velocity were observed between WH1 and WH2 athletes. Given $TT_{mean}$'s representation of 13 to 21% of the total sprint duration and is influenced by athletes' velocities, it was decided to exclude this parameter from the performance metrics considered for hierarchical clustering.

## Principal component analysis and hierarchical clustering

The PCA revealed two principal components explaining together 64.35% of the overall variance. The first component (PC1) accounts for 54.30% of the variance and is linked to velocity parameters as indicated in Table 4. The second component (PC2) explains 10.05% of the variance and is related to temporal and propulsion technique parameters.

Three distinct clusters were identified through hierarchical clustering analysis on PCA components. The three clusters are represented in Fig 3.

Regarding PC1, significant differences were noted across all clusters ($p < 0.001$). Regarding PC2, significant differences were noted for the cluster 1 ($p < 0.001$) and cluster 3 ($p = 0.002$).

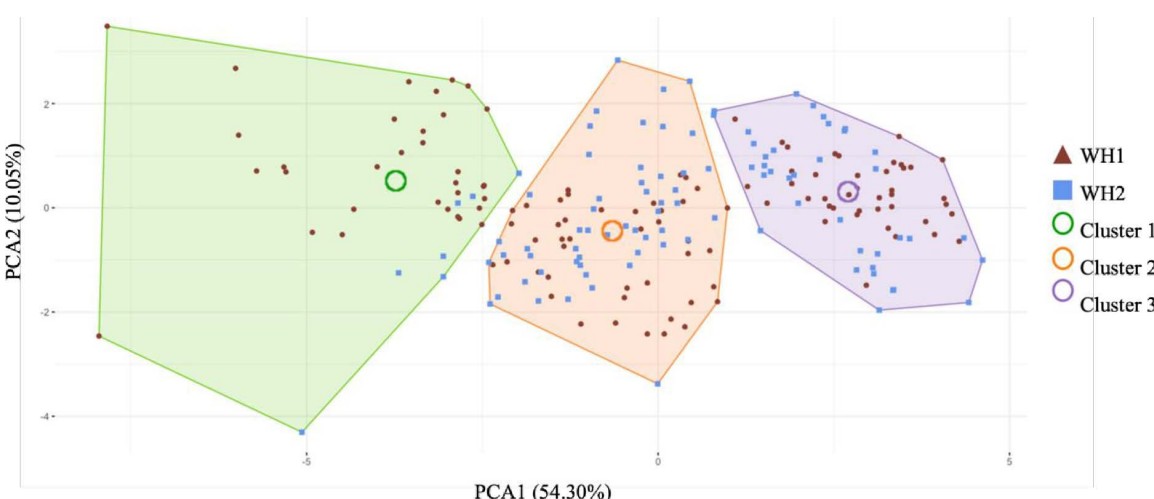

**Fig 3. Clustering analysis in each variable on each PCA component.** With colored polygons: each cluster, colored dot: observations of each athlete in dark red for WH1 and blue for WH2.

**Table 4. Results of the principal component analysis and coordinate of each variable in each PCA components as well as percentage of variance.**

|  | Principal component 1 (PC1) | Principal component 2 (PC2) |
|---|---|---|
| Variance (%) | 54.30 | 12.05 |
| Forward propulsion |  |  |
| $DP_{mean}$ | -0.557 | 0.703 |
| $ST_{mean}$ | -0.791 | 0.468 |
| $V_{max}$ | 0.850 | 0.293 |
| $V_{peak}$ | 0.599 | 0.003 |
| $A_{mean}$ | 0.727 | -0.047 |
| $D_{mean}$ | 0.772 | -0.466 |
| Backward propulsion |  |  |
| $DP_{mean}$ | -0.376 | -0.242 |
| $ST_{mean}$ | -0.854 | 0.043 |
| $V_{max}$ | 0.870 | 0.309 |
| $V_{mean}$ | 0.822 | 0.436 |
| $V_{peak}$ | 0.693 | 0.002 |
| $D_{mean}$ | 0.768 | 0.288 |

But no significant difference was observed for cluster 2 meaning that PC2 contribute contributes slightly less to the distinction between clusters than the PC1. These significant differences make it possible to validate the discrimination of the clusters on the principal components identified. The percentage distribution of classifications by cluster is available in Table 5.

Cluster 1 is predominantly composed of WH1 athletes at 82.50% compared to 17.50% WH2. In contrast, the distributions of clusters 2 and 3 are much more balanced, with 45.26% WH1 and 54.74% WH2 for cluster 2, and 52.56% WH1 and 47.44% WH2 in cluster 3.

**Table 5. Contingency table representing the percentages of WH1 and WH2 in each cluster (in %).**

|  | WH1 | WH2 |
|---|---|---|
| Cluster 1 | 82.50 | 17.50 |
| Cluster 2 | 45.26 | 54.74 |
| Cluster 3 | 52.56 | 47.44 |

## Discussion

To our knowledge, the analysis presented in this article represents the first exploration within the realm of wheelchair badminton. Its main objective of this study was to explore the utility of the temporal and kinematic data link to propulsion technique and athlete performance to contribute evidence-based classification in wheelchair badminton. Differences in the parameters measured were observed between the two classifications in a way that indicates a longer braking time for WH1 compared to WH2, as well as overall lower performance in terms of sprint times and velocity parameters. Contrary to our initial hypothesis, hierarchical clustering analysis did not align with current classifications, revealing three distinct clusters instead of two. This method is an attempt to allow the current classification system to evolve towards an evidence-based classification, by including objective scientific measurements of the performance of the athletes considered and their propulsion technique. Future research is needed to implement this method and its database.

**Comparison of data between WH1 and WH2 athletes.** Regarding propulsion technique parameters, WH2 athletes exhibit a lower deceleration phase compared to those WH1, regardless of the direction of propulsion considered. This is in line with the initial hypothesis. However, the propulsion phase shows no significant differences between the two classes for both propulsion directions. It appears that athletes classified WH2 modify their propulsion technique by reducing braking phases. This observation could be attributed to the superior abdominal capabilities of WH2 athletes, allowing them to lean further forward or backward on the wheelchair compared to WH1 athletes, thereby enabling more effective braking. Indeed, a recent study of Garner & Ricard [27], had shown that athletes with lower trunk functional capacity exhibited higher angular impulse and trunk extension angles during braking [27]. In addition to the challenges related to trunk mobility, individuals without abdominal strength need to stabilize themselves with one hand during braking phases to avoid tipping forward or backward. This prevents them from fully braking with both hands.

In forward propulsion, WH1 athletes exhibit longer sprint times, lower maximum and peak velocities, and reduced acceleration and deceleration compared to WH2 athletes. Similar results are observed in backward propulsion, except for acceleration, where no significant difference is found between the WH1 and WH2 groups. Also, the transition time for WH1 athletes is greater than for WH2 in backward propulsion, while their velocities are decreasing. These results confirm the initial hypothesis and can be attributed to functional abilities, particularly abdominal strength, which plays a crucial role in trunk mobility. Indeed, trunk movements and stability directly influence the functional performance of athletes, as they are essential mechanisms for generating propulsion force [28–30]. Moreover, a more severe impairment leads to decreased postural stability and propulsion capabilities [15]. Thus, WH1 athletes, who have greater impairment of the trunk and abdominal muscles, experience more pronounced negative effects on their wheelchair propulsion performance compared to WH2 athletes.

Research on wheelchair sports performance among reveals significant differences based on athlete classification, particularly in sports like rugby and basketball [16,17,31–33]. Higher-classified athletes tend to have better aerobic and anaerobic capacity, upper limb strength, and sport-specific skills, with parameters like oxygen consumption, sprint velocity, and game

efficiency favoring high-point players [34,35]. Additionally, higher-classified athletes generally show better shoulder strength, aerobic, and anaerobic capacity, underlining the importance of both biomechanical and physiological factors in training [33]. These results are consistent with ours and highlight the importance of using such parameters in the athlete classification process.

## Hierarchical clustering analysis

Cluster analysis on the principal components of the PCA revealed three clusters, contrary to the initial hypothesis of two expected clusters. According to the Mann-Whitney post-hoc test, only PC1 contributes significantly to distinguishing between clusters. PC1 primarily represents velocity data of athletes, which appears to be the most discriminative factor in classifying athletes into three clusters. This finding suggests that the data from this study do not adequately classify athletes according to the two existing classes of wheelchair badminton. However, while the cluster analysis did not reveal two clusters, it revealed three. The distribution of athletes across the different clusters varies. In particular, the first cluster is predominantly composed of WH1 athletes (65.75%), while clusters 2 and 3 have much more balanced distributions between WH1 and WH2. To understand this distribution, the number of years of practice was studied as an explanatory factor. Athletes are classified as "less experienced" if they have less than 5 years of practice, as "experienced" if they have between 5 and less than 10 years of experience, and as "very experienced" if they have 10 years or more of practice. Following this grouping, the proportions were determined in each cluster (see Table 6).

Based on the findings from Table 6, it is evident that cluster 1 predominantly consists of athletes labeled as "less experienced" (57.53%), while cluster 3 almost exclusively comprises "highly experienced" athletes. Additionally, cluster 2 presents a blend of athletes classified as "less experienced" (46.97%) and "experienced" (50.00%). Consequently, years of experience emerge as a possible explanatory factor for the observed cluster patterns and the distribution of WH1 and WH2 athletes. These findings suggest that years of experience may exert an influence on athletes' performance and their propulsion technique.

Recent studies have explored the use of clustering methods, like in this article, for classification in wheelchair sports. Marszałek et al. [36] found significant correlations between functional classes and anaerobic power, suggesting a valid division of wheelchair basketball players into four different classes. Van der Slikke et al. [37] used inertial sensors to measure wheelchair mobility performance, revealing only two performance-based clusters in wheelchair basketball, suggesting a potential reduction in classification groups. Connick et al. [9] employed cluster analysis of isometric strength tests for wheelchair racing, producing four clusters that better reflected activity limitations compared to the current classification system. These studies and the results of this article highlight the potential of data-driven clustering approaches to enhance classification systems and inform coaching strategies in wheelchair sports.

However, the results of this study need to be interpreted with caution. Indeed, in this article, performance parameters were selected with propulsion technique parameters for clustering analysis due to their rapid measurement and ease of acquisition. However, it is important

**Table 6. Contingency table representing the percentages of beginners, intermediates and advanced in each cluster (in %).**

|  | Less experienced | Experienced | Very experienced |
|---|---|---|---|
| Cluster 1 | 57.53 | 42.47 | 0.00 |
| Cluster 2 | 46.97 | 50.00 | 3.03 |
| Cluster 3 | 4.05 | 59.46 | 36.49 |

to acknowledge that an athlete's performance can vary for various reasons, both intrinsic and extrinsic. Taking these considerations into account, along with the results obtained in this study, which did not allow for the identification of the current classification, the decision to use performance parameters in the clustering analysis raises questions. Thus, the results of this article point more to the need to improve the current methodology employed, than to a revision of the current classification. One conceivable approach would be to incorporate functional parameters, such as trunk or upper limb mobility. In wheelchair basketball, for example, functional classification assesses the trunk and upper limb capacities of players, reflecting their ability to perform various actions on the court. The validity of this approach has been demonstrated in various fields, including physiology, biomechanics, and game performance [2,38,39]. Integrating similar objective biomechanical measures could strengthen the clustering methodology of this study and lead to a more precise classification based on tangible data. This evolution could enhance the classification process of wheelchair badminton players, or even prompt its revision if the results justify it. Moreover, such an approach would refine and scientifically reinforce the classification, bringing it closer to the concept of evidence-based classification.

## Limitations

The primary limitation of this study is the absence of kinetic data. Including such data could have provided a more comprehensive assessment of athlete performance and further validated the established clusters. Future research could address this by incorporating additional kinetic measurements to enhance the robustness of the findings.

Furthermore, the instruction given to athletes during the tests was to align the large wheelchair wheel with the start/finish lines to validate the sprint. Despite the particular attention paid to adhering to this instruction, it is possible that athletes did not consistently comply with it throughout the entire 1-minute test duration. Considering this, it is conceivable that the distance covered by each athlete may vary, potentially influencing the velocity results. An improvement could involve employing photoelectric cells at the start and finish lines, signaled by an audible tone upon passage. Athletes would then be instructed to trigger the cell at the end of each sprint.

## Perspectives

Regarding the characterization of WH1 and WH2 athletes, the results obtained in this article could serve as a performance database for coaches and athletes. Indeed, this could enable them to assess their position relative to a global average level, with the potential for individualization and strategic direction in athlete training.

A concrete perspective for the clustering results presented in this article would be to propose to the BWF to integrate more tests based on the evaluation of athletes' functional capacities and objective biomechanical measurements into the existing classification process. Indeed, with the aim of objectifying the classification process using rapid tests and non-invasive measurements, the current classification could become more representative of athletes' abilities, based on the concept of evidence-based classification. In addition to velocity measurements, it would be interesting to add the evaluation of the volume of action of athletes with measurements of trunk kinematics to the classification process.

## Conclusion

To conclude, this article highlights the following points:

- WH1 athletes have a longer braking phase than WH2 athletes.

- Overall, WH1 athletes perform less well than WH2 athletes regardless of the direction of propulsion.

- The temporal and kinematic data in this study did not allow for the identification of the current classification, as three clusters emerged instead of two.

Thus, the results of this study could suggest a potential evolution of the current classification towards three distinct classes of wheelchair badminton players. However, these findings should be interpreted with caution, given that the included performance parameters can be influenced by numerous factors, potentially undermining the robustness of the clustering methodology employed.

This study highlights the need to strengthen the current classification process in wheelchair badminton. To align with IPC guidelines and enhance the methodology employed in this study, integrating functional trunk capacity data could prove to be beneficial.

## Supporting information

**S1 Table. Processed data set used in this article.**
(XLSX)

## Acknowledgments

The authors thank the participants in the study as well as the laboratory of the J-AP2S and the Fédération Française de Badminton.

## Author contributions

**Conceptualization:** Arnaud Faupin.

**Data curation:** Ilona Alberca, Félix Chénier, Florian Brassart, Mélanie Baconnais, Bryan Le Toquin, Imad Hamri.

**Formal analysis:** Mélanie Baconnais, Bryan Le Toquin, Imad Hamri.

**Funding acquisition:** Arnaud Faupin.

**Investigation:** Ilona Alberca.

**Methodology:** Bruno Watier, Mélanie Baconnais, Bryan Le Toquin, Imad Hamri.

**Resources:** Florian Brassart, Jean-Marc Vallier.

**Software:** Mélanie Baconnais, Bryan Le Toquin, Imad Hamri.

**Supervision:** Bruno Watier, Félix Chénier, Arnaud Faupin.

**Validation:** Bruno Watier, Félix Chénier, Mélanie Baconnais, Bryan Le Toquin, Imad Hamri, Jean-Marc Vallier, Arnaud Faupin.

**Visualization:** Mélanie Baconnais, Bryan Le Toquin, Imad Hamri.

**Writing – original draft:** Ilona Alberca.

**Writing – review & editing:** Ilona Alberca, Bruno Watier, Félix Chénier, Mélanie Baconnais, Bryan Le Toquin, Imad Hamri, Arnaud Faupin.

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
