## [Decision Letter · Decision Letter 0]

26 Aug 2024

PONE-D-24-26194Trying to use spatiotemporal parameters for the classification in wheelchair badminton.PLOS ONE

Dear Dr. Alberca,

Thank you for submitting your manuscript to PLOS ONE. After careful consideration, we feel that it has merit but does not fully meet PLOS ONE’s publication criteria as it currently stands. Therefore, we invite you to submit a revised version of the manuscript that addresses the points raised during the review process.

**ACADEMIC EDITOR:** Reviewers have now commented on your paper. You will see that they are advising that you revise your manuscript. If you are prepared to undertake the work required, I would be pleased to consider a revised version.

We look forward to receiving your revised manuscript.

Kind regards,

Javier Abián-Vicén, Ph.D.

Academic Editor

PLOS ONE

Journal Requirements:

2. Thank you for stating the following financial disclosure: "This work was supported by a French government grant, managed by the Agence Nationale de la Recherche (ANR) under the "France 2030" program, reference ANR-19-STHP-0005."  

3. Acknowledgments Section: Move New Information to the Financial Disclosure:

Thank you for stating the following in the Acknowledgments Section of your manuscript: The authors thank the participants in the study as well as the laboratory of the IAPS and the Fédération Française de Badminton.

375 The authors declare no conflict of interest. This work was supported by a French government grant, managed by the Agence Nationale de la Recherche (ANR) under the "France 2030" program, reference ANR-19-STHP-0005. 

Please remove any funding-related text from the manuscript and let us know how you would like to update your Funding Statement. Currently, your Funding Statement reads as follows: "This work was supported by a French government grant, managed by the Agence Nationale de la Recherche (ANR) under the "France 2030" program, reference ANR-19-STHP-0005."

Reviewers' comments:

Reviewer's Responses to Questions

**Comments to the Author**

1. Is the manuscript technically sound, and do the data support the conclusions?

Reviewer #1: Yes

Reviewer #2: Yes

2. Has the statistical analysis been performed appropriately and rigorously? 

Reviewer #1: Yes

Reviewer #2: Yes

3. Have the authors made all data underlying the findings in their manuscript fully available?

Reviewer #1: Yes

Reviewer #2: Yes

4. Is the manuscript presented in an intelligible fashion and written in standard English?

Reviewer #1: Yes

Reviewer #2: Yes

5. Review Comments to the Author

Reviewer #1: Dear Authors,

All comments and recommendation you will find in notes in the PDF file.

I hope they are clear for you.

please, highlight all changes in the next version of the manuscript (e.g. blue fond of new parts in the text).

Kind regards,

Reviewer

Reviewer #2: The paper presents interesting topic of wheelchair badminton athletes classification based on spatiotemporal parameters. The article is well-written, however, the following remarks should be considered:

• The motivation of the study should be indicated more clearly.

• Please correct dot in the statement “In the initial analysis, significant disparities in velocity were observed between WH1 and WH2 athletes that may have large influence on TTmean”.

• Adding more clustering methods (like K-Means and Density-based Clustering) would definitely improve the study.

6. PLOS authors have the option to publish the peer review history of their article (what does this mean? ). If published, this will include your full peer review and any attached files.

**Do you want your identity to be public for this peer review?** For information about this choice, including consent withdrawal, please see our Privacy Policy .

Reviewer #1: **Yes: ** Jolanta Marszałek

Reviewer #2: No

---

## [Author Response · Author response to Decision Letter 1]

12 Sep 2024

Dear Jolanta Marszałek,

Thank you very much for your thorough and precise feedback, which has greatly contributed to improving the quality of the article. We have carefully incorporated all your comments and have addressed them in the section below.

Once again, thank you for your valuable work.

Kind regards,

On behalf the authors, Ilona Alberca

Dear Reviewer,

Thank you for your positive feedback and for recognizing the relevance of the topic. We will carefully consider and address each of the remarks you have provided to further improve the paper. Thank you.

Kind regards,

On behalf the authors, Ilona Alberca

---

## [Decision Letter · Decision Letter 1]

3 Dec 2024

Trying to use kinematics and temporals parameters for the classification in wheelchair badminton.

PONE-D-24-26194R1

Dear Dr. Alberca,

We’re pleased to inform you that your manuscript has been judged scientifically suitable for publication and will be formally accepted for publication once it meets all outstanding technical requirements.

Kind regards,

Javier Abián-Vicén, Ph.D.

Academic Editor

PLOS ONE

Reviewers' comments:

Reviewer's Responses to Questions

**Comments to the Author**

1. If the authors have adequately addressed your comments raised in a previous round of review and you feel that this manuscript is now acceptable for publication, you may indicate that here to bypass the “Comments to the Author” section, enter your conflict of interest statement in the “Confidential to Editor” section, and submit your "Accept" recommendation.

Reviewer #3: All comments have been addressed

Reviewer #4: All comments have been addressed

2. Is the manuscript technically sound, and do the data support the conclusions?

Reviewer #3: Yes

Reviewer #4: Yes

3. Has the statistical analysis been performed appropriately and rigorously? 

Reviewer #3: Yes

Reviewer #4: Yes

4. Have the authors made all data underlying the findings in their manuscript fully available?

Reviewer #3: Yes

Reviewer #4: Yes

5. Is the manuscript presented in an intelligible fashion and written in standard English?

Reviewer #3: Yes

Reviewer #4: Yes

6. Review Comments to the Author

Reviewer #3: The authors have responded satisfactorily to the comments made by the reviewers. Therefore, my recommendation is that the paper be accepted in the current form.

Reviewer #4: I think you have been addressed all comments and the manuscript meets the criteria established by Plos One. Congratulations for your work.

7. PLOS authors have the option to publish the peer review history of their article (what does this mean? ). If published, this will include your full peer review and any attached files.

**Do you want your identity to be public for this peer review?** For information about this choice, including consent withdrawal, please see our Privacy Policy .

Reviewer #3: No

Reviewer #4: No

---

## [Editor Report · Acceptance letter]

PONE-D-24-26194R1

PLOS ONE

Dear Dr. Alberca,

I'm pleased to inform you that your manuscript has been deemed suitable for publication in PLOS ONE. Congratulations! Your manuscript is now being handed over to our production team.

Kind regards,

on behalf of

Dr. Javier Abián-Vicén

Academic Editor

PLOS ONE